# Effect of *Rumex nervosus* Leaf Powder on the Breast Meat Quality, Carcass Traits, and Performance Indices of *Eimeria tenella* Oocyst-Infected Broiler Chickens

**DOI:** 10.3390/ani11061551

**Published:** 2021-05-26

**Authors:** Mohammed M. Qaid, Saud I. Al-Mufarrej, Mahmoud M. Azzam, Maged A. Al-Garadi, Abdulmohsen H. Alqhtani, Esam H. Fazea, Gamaleldin M. Suliman, Ibrahim A. Alhidary

**Affiliations:** 1Animal Production Department, College of Food and Agriculture Sciences, King Saud University, Riyadh 11451, Saudi Arabia; salmufarrej@ksu.edu.sa (S.I.A.-M.); mazzam@ksu.edu.sa (M.M.A.); malgaradi@ksu.edu.sa (M.A.A.-G.); ahalqahtani@ksu.edu.sa (A.H.A.); essamhazaa42@gmail.com (E.H.F.); gsuliman@ksu.edu.sa (G.M.S.); ialhidary@ksu.edu.sa (I.A.A.); 2Faculty of Veterinary Medicine, Thamar University, Dhamar 13020, Yemen; 3Poultry Production Department, Faculty of Agriculture, Mansoura University, Mansoura 35516, Egypt; 4Department of Meat Production, Faculty of Animal Production, University of Khartoum, Khartoum North 13314, Sudan

**Keywords:** breast, broiler, carcass, meat quality, *Rumex nervosus*

## Abstract

**Simple Summary:**

Recent research has demonstrated the beneficial effects associated with the addition of *Rumex nervosus* leaves (RNL) to the diet of broiler chickens. Coccidiosis, a disease caused by Eimeria, causes great economic losses in the poultry industry worldwide due to its prevalence and associated management issues. There has been some evidence that RNL can be fed to broiler chicks to prevent infections of the protozoa *Eimeria tenella*; however, there is no or insufficient evidence on the effects of RNL on growth performance and meat quality of broilers infected with *Eimeria tenella*. This research aimed to evaluate the effects of RNL on performance indices, carcass traits, and meat quality of broilers infected with *Eimeria tenella*-fed diets supplemented with RNL as a natural anticoccidial drug replacement. RNL had a positive effect on performance indices, meat quality, and some carcass traits in broilers infected with *Eimeria tenella*, according to the findings. Furthermore, 1 g of RNL led to a better quality of meat than other RNL doses concerning enhanced holding capacity of water, drip loss, myofibrillar fragmentation index (MFI), springiness, and tenderness. At higher doses (3 g of RNL), there was a noticeable decrease in the meat’s final pH, MFI, and springiness.

**Abstract:**

This work aimed to assess the effect of using the RNL on performance indices, carcass trait, and meat quality of broiler chicken infected with *Eimeria tenella* compared with commercially used anticoccidials salinomycin. Moreover, we compare these selected variables between infected medicated groups and a non-infected unmedicated group (UUT) and an infected unmedicated group (IUT). A total of 150 1-day-old Ross 308 mixed-sex broilers were divided into 6 groups. Birds in groups 1, 2, and 3 were fed basic diets supplemented with 1, 3, and 5 g of RNL/kg diet, respectively. Group 4 received a basic diet with 66 mg of salinomycin. The control groups (5 and 6) were given a basic diet with no RNL or salinomycin added. All groups except the negative controls were challenged with *Eimeria tenella* at 21 days old. Birds in RNL groups outperformed those in the IUT group in performance indices, and they have a similar effect to the salinomycin group. Enhancement-infected birds with RNL affected some chickens’ carcass traits. Drip loss, water-holding capacity, and meat tenderness were improved by RNL inclusion (1 g) in the diet. In conclusion, the meat of infected birds receiving 1 g RNL had increased quality attributes, with preferable tenderness and springiness when compared to the IUT group. RNL could therefore also be considered a promising non-conventional feed source in the future. Further research is needed to optimize the use of RNL to improve broilers production and meat quality in both infected and non-infected conditions.

## 1. Introduction

The poultry industry remains a major source of high-quality proteins, vitamins, and essential micronutrients for human nutrition. Increasing competition for poultry meat reflects its high nutritional value and satisfactory price compared to that of other meat [1]. The global poultry industry accounted for 37% of meat produced in 2017 and is projected to produce approximately 331 million tons of meat in 2028 [2]. This tremendous growth has been correlated to the subtherapeutic use of antibiotics as growth promoters (AGPs). An increase in meat production to meet the growing demand for poultry meat has also led to increased efforts to improve the quality of meat. The dressing percentage (DP) is significant because it determines the amount of material available for sale. Consumers prefer products with a high muscle content and a low-fat content [3].

Broiler mass production has already been achieved, and the focus is now on improving meat quality by modifying various aspects of broiler meat. According to [4], the most perceptible and important meat features that influence consumers’ initial and final quality judgments before and after purchasing a meat product are appearance, texture, juiciness, tenderness, wateriness, firmness, flavor, and odor. Quantifiable meat properties such as pH, shelf life, drip loss, shear force, cook loss, collagen content, cohesiveness, protein solubility, water-holding capacity, and fat-binding capacity are critical for processors involved in the production of value-added meat products [5]. The diet of broiler chickens has a significant impact on the quality and safety of poultry meat [6]. Following slaughter, biochemical changes occur, resulting in the conversion of muscle to meat and determining the final meat quality [7]. The temperature of the postmortem carcass has a significant impact on rigor mortis. The physicochemical changes in PSE muscles are caused by postmortem glycolysis, pH, and temperature [8]. Large differences in meat quality and rigor mortis completion rates among chickens could be attributed to genetic variation. Heritability estimates for meat quality traits in broilers are extremely high (0.35–0.81), making genetic selection the most effective tool for improving broiler meat quality [9]. Coccidiosis harms broiler chicken growth rate and meat quality, resulting in economic losses [10]. They found that natural food additives such as nano-curcumin can improve the organoleptic properties of poultry meat during shelf life.

Conventional feed supplements and additives are an essential component of livestock feed that have been used to reduce pathogens, improve zootechnical indices, optimize feed efficiency, and enhance animal production [11]. European legislation allows coccidiostats to be used as feed additives for specific animal species [12]. The European Union imposed a total ban on antimicrobial growth promoters in 2006, which meant that antimicrobials other than coccidiostats were no longer permitted as feed additives in the poultry industry [13]. The contribution of poultry meat and eggs to the acceptable daily intake of each coccidiostat is less than 1%, indicating a low direct risk to public health [14]. In some countries, since the late 1990s, conventional coccidiostats have been linked to undesirable residue levels in meat and eggs [15]. This has resulted in increased monitoring and research of coccidiostat residues in food. Gastrointestinal problems caused by *Eimeria* spp. controlled by coccidiostat agents incorporated into the feed are some of the leading causes of poor performance in the poultry industry. Although ionophore coccidiostats are widely used in the modern poultry industry, non-traditional feed resources or natural additives such as shrubs or their parts may be a desirable replacement to facilitate the discontinuation of continuous use of in-feed ionophore coccidiostats [16]. For some anticoccidial drugs, such as diclazuril, the residual activity of anticoccidial drugs in chickens after withdrawal of medicated feeds for up to 3 days before *Eimeria tenella* inoculation was unique. On the other hand, salinomycin, monensin, amprolium, halofuginone, nicarbazin, lasalocid, and robenidine showed no residual anticoccidial activity [17].

Plants of the genus *Rumex* are used in traditional medicines and have been investigated for phytochemicals and pharmacological activities [18,19], including the shrub *Rumex nervosus* (Othrob), belonging to the genus *Rumex* and family Polygonaceae [20]. *Rumex nervosus* extract can eliminate or scavenge free radical compounds and protects cells from lipid peroxidation [21,22,23]. Researchers have also shown that due to its antimicrobial and anticoccidial properties, *Rumex nervosus* leaves (RNL) can be used as a performance enhancer, and it also protects the intestinal mucosa by increasing the levels of antioxidant enzymes [23,24]. Data on broiler growth performance, serum biochemical indices, cecal microbiota, and duodenal histomorphology fed RNL [25], as well as anticoccidial efficacy [26], have recently been published. However, the impacts of RNL as an anticoccidial drug have not been studied previously. In addition, their effects are being investigated before marketing to determine if they have a positive role on performance indices, as well as improvement of meat quality and carcass traits of infected broilers with *Eimeria tenella*. In addition, not much is known about the dietary RNL in poultry-produced meat infected with *Eimeria tenella*. Thus, the goal of our research was to learn about the effect of RNL as an anticoccidial drug within broiler diets after being experimentally infected by *Eimeria tenella* on carcass traits, growth performance, and meat quality when compared to a commercial drug. Moreover, we compare these selected variables between infected/drug-treated groups and a non-infected/untreated group (UUT) and infected/untreated group (IUT).

## 2. Materials and Methods

This research is a continuation of an already published study. The anticoccidial indices evaluation of RNL, such as bloody diarrhea, fecal oocysts number, lesion scores, and survival rate, were determined in that study [26]. As a result, the evaluation of anticoccidial indices is not covered in this study. The growth performance data of infected birds were collected here between the ages of 21 and 34 days (1–14 days post-infection “dpi”). All treatments were subjected to meat quality and carcass parameters’ measurements on the final day of the experiment, which was day 34 of age (14 dpi).

The precautions were taken to prevent adventitious infection before 21 days: this study extended to a previous study done by feeding broilers different levels of RN herb and salinomycin and published this work in [25]. Since the beginning of the trial period (from day 1 of age), all experimental groups have been subjected to the same dietary medication additions. We detected that the RNL has effectiveness on Emeria. In addition, salinomycin is a standard drug.

The direct smears from fecal material before this age were taken to detect the presence of oocysts under a microscope and the results were negative and did not detect Emeria Oocyst. In addition, direct blood smears were performed, and Eosinophil levels were found to be within the normal range. While after infection, Eosinophils were found with a high number in all infected groups compared with the non-infected group.

### 2.1. Ethical Approval and Consent to Participate


This study was conducted in compliance with the guidelines of the National Animal Care and Use Committee of Saudi Arabia for ethical and scientific research use of animals. The procedures and protocols included in this study were approved by the Animal Care and Use Ethics Committee of the King Saud University (KSU-SE-20-44).

### 2.2. Birds’ Housing and Management

One-day-old mixed-sex Ross 308 broiler chicks (*n* = 150; 75 ♀ and 75 ♂) were obtained from Al Khomasia hatchery, Riyadh, Saudi Arabia, following immunization against Newcastle disease and infectious bronchitis. Chicks were randomly assigned to six experimental groups (five replicates for each treatment, five chicks per replicate) and raised in cages at the Poultry Research Unit affiliated with King Saud University, Riyadh, Saudi Arabia. The chicks were housed in an environmentally controlled heated battery room fitted with a 24 h lighting schedule under similar zoo-hygienic and managerial conditions as previously reported. Broiler chickens were infected with *Eimeria tenella* at 21 days of age, according to [23,27]. At the age of 21 days, all birds except the negative control group were artificially infected with 4 × 10^4^ sporulated oocysts of *Eimeria tenella*/mL/bird. For the entire study period, drinking water and experimental feed were provided *ad libitum*.

### 2.3. Dietary Experimental Treatments

Starter (days 1–21) and finisher (days 22–34) commercial diets were mixed and formulated in a mashed form based on diets of corn-SBM, and the content and chemical analyses of nutrients were as suggested in [28] (Table 1). Upon arrival, birds were randomly assigned to one of the following six treatment groups: UUT = negative control group, unmedicated diet, unchallenged *Eimeria tenella* Oocysts chicks, IUT = positive control group, unmedicated diet + coccidial challenge, SAL = medicated diet with 0.066 g of salinomycin sodium/kg diet + coccidial challenge (SAL was used as the reference anticoccidial drug), and 1 g RNL, 3 g RNL, and 5 g RNL = 1, 3, and 5 g RNL powder/kg diet respectively, + coccidial challenge. According to [29,30], active ingredients of anticoccidial drugs (RNL and SAL) were not included in the nutrient matrix but were mixed on top of the basal broiler fed due to its use in the lowest concentrations ranging between 0.1 and 0.5 percent for RNL and 0.0066 percent for SAL.

### 2.4. Collection and Compositions of Rumex nervosus Leaf Powder

RNL was harvested, exsiccatae were registered, air-dried, ground, and nutritional value was determined using proximate analysis, as described in [26]. Furthermore, as described in [26], the high-performance liquid chromatography (HPLC) analysis and gas chromatography-mass spectrometry (GC-MS) test were used for detecting, identifying, and separating chemical compounds in the RN extract mixture.

### 2.5. Growth Performance Indices

To assess the birds’ performance parameters, all birds were weighed separately, and the body weight was determined at 1, 7, and 14 dpi per replicate. Body weight gain (BWG) was calculated by subtracting the initial body weight from the final body weight of birds. The feed consumption (FI) of birds in each pen was measured at 1, 7, and 14 dpi by subtracting the residual feed weight from the feed provided during selected times. Feed efficiency (FE) was calculated as BWG (g) divided by FI (g). The European production efficiency factor (PEF) and performance index (PI) were calculated as follows:(1)Production efficiency factor (PEF)=(Livability×Live body weight (Kg))÷(Age in days×FCR))×100
(2)Performance index (PI)=(Total weight gain÷Total FCR)×100

### 2.6. Slaughter Composition Variables

At the age of 34 days, a random sample of five birds (*n* = 5, one bird/pen/group) from each group were selected and weighed. For 10 h pre-slaughter, birds were starved and then weighed and slaughtered in strict compliance with the animal welfare legislation of the slaughtering process. After bleeding (1.5–2 min following incision of the jugular veins), birds were defeathered and eviscerated. After removing the feathers, head, and shanks, the remaining carcasses were dissected. Live weight (LW) and dressing weight (DW) (inclusion of the giblets (liver, heart, and gizzard) neck, and abdominal fat, without feathers, head, and shanks) were recorded to calculate dressing yields (DY) = (dressing weight/live weight) × 100. Hearts, livers, proventriculi, gizzards, lymphoid organs (spleen, bursa of Fabricius, and thymus), breasts (including pectoralis major and pectoralis minor), legs (including thigh and drumstick), and pancreases were weighed. These weights were then expressed relative to dressing weight; for example, breast meat yield was estimated as breast meat weight relative to dressing weight (breast meat yield = (breast weight/dressing weight) × 100).

### 2.7. Meat Quality Indicators

To measure the quality of the meat, the right and left sections of the pectoralis major of the breast dissected from each of the male birds selected for slaughter were used. Meat pH and color components were measured at 1 and 24 h post-slaughter. Immediately after pH and color quality assessment, the samples of meat were stored at −20 °C until further analysis. Before WHC, cooking loss (CL), MFI, and texture profile analysis (TPA), the frozen muscles were thawed in the refrigerator for 24 h, at 4 °C.

#### 2.7.1. Meat pH

The pH was evaluated using a microprocessor pH-meter (Model pH 211; Hanna Instruments, Woonsocket, RI, USA). The averages of three pH readings for each carcass were recorded at 1 and 24 h. The core temperature of breast meat was measured 1 h postmortem (PM) using a portable digital thermocouple (EcoScan Temp JKT; Thermo Scientific, Waltham, MA, USA).

#### 2.7.2. Color Measurements

To measure meat color, two pictures were taken at 0 and 24 h postmortem in two different fields of the inner face of the cranial position of the breasts. The lightness (L*), redness (a*), and yellowness (b*) of the CIE scales were assessed using the Chroma meter Konica Minolta CR-400 (Konica Minolta, Tokyo, Japan), as described in [31]. The L*, a*, and b* values were used to evaluate total color change (∆E), Chroma meter saturation index, hue angle, browning index (BI), and whiteness index (WI), according to the methods reported in [32,33], using the colorimeter calibration values L_0_* = 94.18, a_0_* = −0.43, and b_0_* = 3.98, where L_0_, a_0_, and b_0_ represent the initial color. Those converted color indicators provide a more realistic assessment of how the consumer perceives the color of food, according to [34]. The equipment was calibrated with a standard white plate.
(3)$$\Delta E=\sqrt{{\left(\Delta L*\right)}^{2}+{\left(\Delta a*\right)}^{2}{\left(\Delta b*\right)}^{2}}=\Delta E=\sqrt{{\left(L*-L_{0}*\right)}^{2}+{\left(a*-a_{0}*\right)}^{2}{\left(b*-b_{0}*\right)}^{2}}$$

$\mathrm{Chroma}\;\left(\mathrm{Saturation}\;\mathrm{Index}\right)=\sqrt{a^{2}+b^{2}}$.
(4)$$\mathrm{Browning}\;\mathrm{index}\;\left(\mathrm{BI}\right)=\frac{100\times \left(\chi -0.031\right)}{0.17}\;\chi =\frac{a\;+\left(1.75\times L\right)}{\left(5.645\times L\right)+a-\left(3.012\times b\right)}$$(5)$$\mathrm{Whiteness}\;\mathrm{index}\;\left(\mathrm{WI}\right)=100-\sqrt{{\left(100-L*\right)}^{2}+{\;a*}^{2}+{b*}^{2}}$$Hue angle = tan − 1 (b/a)(6)

In the three-dimensional CIE L*a*b* uniform color space, the color coordinates of breast meat are determined by the color brightness parameter L* (whiteness/darkness coordinate), which measures whiteness and ranges from 0 to 100, indicating brightness ranges from black to white, respectively. The chromaticity coordinate a* measures red/green hue (red if a* is positive and green if a* is negative), and b* measures yellow/blue hue (yellow when positive and blue when negative).

#### 2.7.3. Water-Holding Capacity (WHC)

The frozen left side of the major pectorals was used directly after thawing at 4 °C for 24 h to measure WHC according to the compression method using the protocol of [35]. In brief, approximately 2–3 g of the muscle was positioned and pressed between two pieces of filter paper into two acrylic-plastic plates, applying a force of 10 kg for 5 min. The samples were weighed, and the WHC values were calculated using the formula below for the exudate water weight:WHC percent = [(prepressed meat weight − post-pressed meat weight)/prepressed meat weight] × 100(7)

#### 2.7.4. Measurement of Cooking Water Loss (CWL)

Cooked breast meat samples were weighed before and after cooking using a semi-analytical scale (Mettler MP1210; Mettler-Toledo Ltd., Leicester, UK) to determine the percentage of cooking water loss (CWL) [36] using the equation: CWL percentage = [(Initial weight − Cooked weight) ∗ 100]/Initial weight. Samples were placed on a commercial indoor counter top grill (Kalorik GR 28,215; Kalorik, Miami Gardens, FL, USA) and boiled or toasted to an internal temperature of 75 °C to determine CL. To monitor the internal temperature, a thermocouple thermometer probe (EcoScan Temp JKT; Eutech Instruments, Singapore) was inserted into the geometric center of the muscular samples.

#### 2.7.5. Measurement of Myofibril Fragmentation Index

Myofibril fragmentation index (MFI) of the meat samples was measured using the procedure reported in [37] as an indirect measure of calpain activity. Briefly, 4 g of scissor-minced muscles, free of visible connective tissue and external fat, were homogenized for 30 s in a blender containing 40 mL of cold MFI buffer (2 °C). After many washes, the absorbance of the resultant 0.5 mg/mL solution was measured at 540 nm with a spectrophotometer (HACH DR/3000 Spectrophotometer, (Hach, Loveland, CO, USA). The MFI of each sample was computed by multiplying the absorbance at 540 nm by 200.

#### 2.7.6. Warner–Bratzler Shear Force (Tenderness) Measurement

Breast muscle shear force (SF, kg/cm^2^) or tenderness was analyzed using the TA-HD Texture Analyzer (Stable Micro Systems Ltd., Godalming, UK). The cooked samples were cooled at room temperature (22 °C). Five rectangular core steaks (2 × 1 × 1 cm in diameter) from each sample were cut in a longitudinal direction parallel to the muscular fibers using a manual coring device. To measure the SF or tenderness, shear force was determined as the maximum force (kgf/cm^2^) that could be applied vertical to the fibers using a TA-HD Texture Analyzer fitted with a Warner–Bratzler shear barb with a triangular orifice-cutting border. The crosshead speed was set at 200 mm/min. The device was operated at speeds of 2, 2, and 10 mm/s at the pre-test, test, and post-test respectively, from a distance of 15 mm. The SF values were measured from the maximum point of the curve generated.

#### 2.7.7. Texture Profile Analysis (TPA)

The TPA of breast meat was performed using a TA-HD Texture Analyzer (Stable Micro Systems Ltd., Godalming, UK). TPA was determined by scoring cooked muscle fibers parallel to the longitudinal direction using a handheld corer’s aid (1.5 cm high and 2.5 cm in diameter). A cylinder-shaped piston (75 mm in diameter) was used to compress the specimen over two test cycles, compressing the specimen up to 80% of the original height, with a time period of 5 s between cycles. Force time curves of deformation were obtained from the conditions laid down in the texturometer. The velocities used were 2.0, 5.0, and 5.0 mm/s at the pre-test, test, and post-test, respectively. The parameters of hardness, chewiness, springiness, and cohesiveness were measured according to the definitions of [38].

### 2.8. Statistical Analysis

All data were analyzed using GLM procedures in Statistical Analysis System (SAS) software [39]. The six groups were arranged in five replicates in a randomized complete design (CRD). Carcass composition traits, meat physicochemical properties, MFI, CL, and WHC, SF parameters, and texture variables were analyzed using the following one-way ANOVA model:Y_ij_ = μ + T_i_+ e_ij_(8)
where Y_ij_ is the value of any observation, μ is the population mean of the measurements, T_i_ is the effect of the *i*th treatment, and e_ij_ is the random error that is normally distributed with the mean zero and variance σ^2^ε_ij_~N (0, σ^2^). Data were expressed as means ± standard error of the mean (SEM), and the means were compared using Duncan’s multiple range test at a level of significance of *p* < 0.05.

## 3. Results

According to [26], coccidiosis was present in broilers after infection with *Eimeria tenella* oocyst. They discovered that RNL powder, at 5 g, has moderate anticoccidial effects and can be used to treat avian coccidiosis in the field. As a result, RNL reduced lesion scores and suppressed oocyst output per gram in bird feces.

### 3.1. Growth Performance Indices

Table 2 displays that the *Eimeria tenella* harmed bird body weight, gain, feed intake, feed conversion ratio, and production efficiency during the first week after infection when compared to an unchallenged group (UUT) (*p* < 0.05). Fortunately, the medicated groups (SAL or RNL) were able to compensate for the growth losses caused by *Eimeria tenella* and then improved all performance parameters in the second week after infection when compared to an infected/unmedicated group (IUT) (*p* < 0.05). As a result, the BWG, FCR, PI, and PEF of broiler chickens were significantly improved (*p* < 0.05) during the entire period following the *Eimeria tenella* oocyst challenge (1–14 dpi) when compared to an infected/unmedicated group (IUT).

Even though the best results were obtained in the UUT group, all growth performance indices improved significantly in the SAL group, followed by the 5 g RNL group when compared to the IUT.

### 3.2. Carcasses and Body Component Variables

The influence of RNL powder on carcass traits (live weight, carcass yield, and carcass composition) of broilers aged 34 days (14 days post-infection (dpi)) is shown in Table 3. The slaughter weights and carcass weights of the broiler chickens did not vary (*p* > 0.05) between the experimental groups. In addition, there was no difference (*p* > 0.05) in the relative weights of the heart, proventriculus, bursa, thymus, leg, and abdominal fat. The relative weights of the liver, gizzard, spleen, breast, and pancreas, on the other hand, differ significantly (*p* < 0.05) between the treatment groups. In addition, the percentages of dressing yield differ significantly (*p* < 0.05) between the treatment groups. The infected/medicated groups and the uninfected/unmedicated group (UUT) achieved the highest dressing values, while the infected/unmedicated group (IUT) had the lowest. The SAL group had the most substantial breast weights, followed by the 5 g RNL group, and the IUT group had the smallest breast weights. The SAL group and 1 g RNL achieved the highest liver relative weight. The affected groups’ liver, pancreas, gizzard, and spleen values increased significantly (*p* < 0.05) more than the UUT group.

### 3.3. Muscle pH and Color

Results of the impact of RNL on physicochemical indicators such as pH, lightness (L*), redness (a*), and yellowness (b*), and other derivative color measurements for the breast muscle of birds at 1 and 24 h postmortem, are presented in Table 4. The core temperature, initial yellowness, final pH, final lightness, final total color change (∆E), and final whiteness index (WI2) of breast meat were significantly different between experimental groups and the controls (*p* < 0.05). On the other hand, initial pH, initial lightness, initial total color change (∆E), and final yellowness did not differ between experimental treatments (*p* > 0.05). In addition, the initial and final redness, hue angle, browning index (BI), and saturation index (Chroma) did not differ between experimental treatments (*p* > 0.05).

The initial temperature 20 min postmortem was significantly different (*p* < 0.0001) between the treatment groups. The 1 g RNL group had the highest initial temperature (28.55 °C), while the UUT treatment group had the lowest (24.89 °C).

The initial pH did not differ (*p* > 0.05) from group to group, whereas the ultimate pH values were considerably different (*p* < 0.05). The 5 g RNL treatment led to the highest final pH values, while the SAL and 3 g RNL treatment exhibited the lowest pH values. In general, the highest pH decrease was observed in the 3 g RNL group between the initial and final pH states.

The 5 g RNL group had a lower initial yellowness (b*) color coordinate, while the IUT group had the highest (*p* = 0.045). All challenged groups displayed reduced final lightness (*p* < 0.01) and final whiteness indices (*p* < 0.0205) compared to the UUT group, in contrast to the final total color change (*p* < 0.01); however, the RNL groups did not differ from the IUT group.

Pectoralis muscles of birds displayed higher (*p <* 0.0001) final lightness (L*) values in the unchallenged control group (UUT), which led to poor color change and greater whiteness indices compared to challenged groups, and vice versa; however, there was no significant difference between the 5 g RNL and UUT groups. This suggested that the challenge of Emeriosis could change the color parameter, particularly the lightness of the breast muscle.

### 3.4. Quality of Meat

Characteristics of the quality of the meat (WHC, CL, DL, MFI, and SF) for bird samples at the age of 34 days (14 dpi) are shown in Table 5. The WHC, CL, DL, MFI, and SF parameters of breast meat samples differed (*p* < 0.05) between treatments. The experimental treatments significantly affect WHC: the WHC did have a decrease with an increase in RNL treatment level. This may indicate a decrease in meat juiciness with an increase in the RNL level. The most favorable values for CL, DL, and MFI were obtained with 3 g RNL, 1 g RNL, and SAL treatment groups, respectively. Cooking loss analysis indicated that bird breasts that received the anticoccidial SAL or 1 g RNL had a higher rate of shrinkage when cooking than those that received other treatments (*p* < 0.001). The most convenient CL value (20.83%) was obtained in the 3 g RNL group, followed by IUT, UUT, and 5 g RNL groups, which were lower than the values of the 1 g RNL and SAL groups. The most convenient DL values (0.91%, 0.99%, and 1.01%) were obtained in the 1 g RNL, IUT, and SAL groups respectively, followed by UUT (1.55%), then 3 g RNL (2.04%) and 5 g RNL (2.14%) groups. The lowest shear force (0.86 kg), indicating the most tenderness, was observed in the 1 g RNL treatment, followed by the UUT group (1.32 kg). The 5 g RNL treatment group showed the highest value of shear force (1.76 kg), which is a clear sign of tough meat.

### 3.5. Texture Analysis

Breast meat TPA quality from broilers supplemented with RNL and challenged with Emeriosis is shown in Table 6. Differences in the hardness, springiness, and chewiness values were found among experimental treatments (*p* < 0.05). Higher hardness, springiness, cohesiveness, and chewiness values (*p* < 0.05) were observed in the breast muscle of birds given 66 mg of SAL/kg. Although 1 g of RNL/kg led to equal springiness with the SAL and control groups, the higher springiness values (*p* < 0.05) of 1 g of RNL/kg followed by 5 and 3 g of RNL/kg groups indicated that springiness (tenderness) improved with 1 g of RNL/kg in chicken diets. Hardness and chewiness did not change significantly between various concentrations of RNL supplements or even control treatments (*p* > 0.05), but SAL resulted in significant differences compared to experimental treatments (*p* < 0.05). Springiness was lower in the RNL groups compared to controls and SAL groups (*p* < 0.007), except in the 1 g RNL group, which was similar. In this study, the analysis of texture profile found significant differences between treatments, where RNL supplementation led to lower levels of hardness, springiness, cohesiveness, and chewiness compared to the SAL group, but did not differentiate from the controls.

## 4. Discussion

Since the late 1990s, conventional coccidiostats have been linked to undesirable residue levels in meat and eggs in some countries [15]. There is, however, no convincing published evidence to support the claim that residues exist or have caused a problem for consumers. Except for diclazuril, there is no evidence of residual anticoccidial activity in salinomycin or other anticoccidial drugs [17]. According to [14], there is a low direct risk of each coccidiostat residue in poultry meat and eggs to public health (less than 1%). Furthermore, none of the natural products have been tested to see if they, too, cause residues. On the other hand, the effects of *Rumex nervosus* leaf powder on the breast meat quality, carcass traits, and marketing growth performance of *Eimeria tenella* oocyst-infected broiler chickens were first investigated. The RN plant’s effects on meat quality, carcass traits, and growth performance were not studied directly in this study, but rather the plant’s effects as a drug on meat quality, carcass traits, and growth performance during *Eimeria tenella* infection.

The bird was infected with *Eimeria tenella* via oral uptake of sporulated oocysts that invaded the intestinal cecum, damaging the cecal epithelium due to *Eimeria tenella* stage multiplication from endogenously (schizogony and gametogony) to exogenously (sporogony), and eventually, the developed oocysts were released in feces [40]. Researchers are working hard to develop multiple prevention methods to monitor the effects of *Eimeria tenella* to reduce the amount of money spent on repairing the damage caused in the poultry farms [23,41]. The consequences of *Eimeria tenella* damage ranged from localized intestinal tissue degeneration to death in the most severe cases [24].

To avoid negative effects on broiler performance, new agents with low cost and minimal side effects against *Eimeria tenella* are needed. Our results indicated that RNL reduced the weight loss accompanying *Eimeria tenella* infection, in agreement with the findings of [23], who reported that RNL extract was capable of decreasing Emeriosis-induced weight loss. In addition, [25] found that body weight improved with dietary supplementation of RNL powder during the pre-starter period, although the marketing weight of broilers was not influenced. 

Our findings support those of [24], where *Eimeria tenella* harmed performance indices, and RNL can mitigate *Eimeria tenella* suffering to some extent but not completely. This is evident by the lack of observed differences in performance indices between medicated groups (SAL or RNL) and IUT groups during the first week after infection. The performance indices did not differ significantly between synthetic and natural anticoccidial products, consistent with the comparative anticoccidial study [30]. In accordance with [42], the poor performance of infected birds, whether medicated or not, as a result of the coccidial challenge is associated with nutrient malabsorption, decreased absorptive surface area, and inflammation. Natural growth promoters such as *Thymus vulgaris* and *Rosmarinus officinalis* may play an important role in alleviating coccidiosis symptoms in poultry [43]. In broilers reared without anticoccidial drugs, the combination of garlic and oregano essential oil (OEO) had a potent anticoccidial effect in vitro as well as a growth-promoting effect [44]. In a dose-dependent manner, graded doses of crude methanol extract of *Garcinia kola* significantly improved BWG in *Eimeria tenella*-infected broiler chicks as a result of decreased fecal oocysts per gram [45]. In broiler chickens, *Artemisia annua* improves performance but has little anticoccidial activity [46]. Shi Ying Zi herbal powder could protect infected chickens from *Eimeria tenella* infection via prophylactic or therapeutic administration, while also improving relative growth rate [29]. When compared to an infected/unmedicated group in the second week after infection, the medicated groups (SAL or RNL) were able to compensate for the growth losses caused by *Eimeria tenella,* and then improved all performance parameters. As a result, although the best results were obtained in the UUT group, the performance indices of broilers in the medicated groups (SAL or RNL, particularly 5 g RNL) were improved when compared to IUT throughout the entire period following *Eimeria tenella* oocyst challenge (1–14 dpi). This could be because anticoccidial supplements, whether natural or synthetic, increased digestive enzyme activity and intestinal absorptive surface area [42]. They found that dietary OEO supplementation was undeniably more effective in terms of FCR. However, as a phytochemical with potential anticoccidial activity, OEO was less effective than the conventional in-feed agent monensin in promoting chicken growth after *Eimeria tenella* infection. Two weeks after *Eimeria tenella* infection, dietary OEO resulted in BWG and FCR that were comparable to the non-infected group [47]. OEO contains important bioactive components such as carvacrol and thymol [48]. The effect of *Eimeria tenella* infection on the last stage of rearing of broilers was supported by the previous studies that refer to the lost weight at this stage [23,26], and here confirm losses in the IUT group.

In our study, there was no substantial change in some carcass characteristics for birds fed a basal diet or diet supplemented with RNL powder. The insignificant difference between RNL and salinomycin on some parameters indicated that RNL had no adverse effect. Although the experimental treatment groups did not affect slaughter and dressing weight, the carcass yield was higher in the infected medicated groups than in the IUT group. Carcass composition, high carcass yield percentage (74.36%), high breast muscle content, and low fat content are indicators of increased consumer desirability and determine the quality of material for sale [5]. Our results are in partial agreement with [49], who did not observe any significant improvement in the slaughter characteristics of broiler chicks fed with a diet complemented with gallic acid (GA). However, it was noted that the inclusion of a graded dietary GA concentration of 100 or 150 mg/kg enhanced the relative percentage of the breast muscle at 42 days old. In addition, [25] reported maximal relative weight of the birds’ breast muscle at 10 days old when fed a diet supplemented with RNL, but no other slaughter variables differed. This discrepancy in the lack of improvement in the breast meat percentage or in the higher pancreas weight percentage may be attributed to the coccidiosis applied in our study. The results of dressing yield obtained in this study agreed with those of [25,49], who reported a nonsignificant effect on dressing yield dependent on RNL powder or its derivatives (GA) in the broiler diet. Carcass characteristics such as dressed yield percent and relative organ weight were higher in both infected and uninfected Madar leaf powder-supplemented treatments, followed by amprolium and un-supplemented treatments [50]. Carcass characteristics of the infected/medicated nano-curcumin 300 mg/kg group improved when compared to the IUT and infected/medicated nano-curcumin 400 mg/kg groups [10].

Here, the increased relative liver and pancreatic weight in the challenged groups compared to the coccidiosis-unchallenged group may be attributed to the increased liver and pancreatic activity after *Eimeria tenella* infection. Changes in hormones (pancreatic insulin, pancreatic glucagon, and glucagon-like immuno-reactants) measured in *Eimeria acervulina* and *Eimeria tenella* infections have shown no direct statistically significant change in pancreas weight, body weight, liver glycogen, or blood glucose [51]. Both parameters, relative weights of breast muscle and live body weight of broilers, were the worst in the IUT group and the best in the UUT group, indicating that *Eimeria tenella* can hurt body weight growth, particularly breast muscle.

At low doses, *Eimeria tenella* infection does not affect meat quality traits [52]. Phytogenic feed additives, on the other hand, had a significant impact on the quality of broilers [53,54] and pigs’ meat [55]. To the best of our knowledge, this is the first report to investigate meat quality of birds fed RNL powder supplements. Most scientific publications in the literature show the in vitro activity of RNL, rather than in vivo effects. New studies have attempted to introduce RNL into the poultry sector as a growth promoter or as an anticoccidial agent; however, the quality of meat was not addressed. Therefore, there are no data available to which we can compare our results on meat quality. However, the data clearly showed that decreased RNL doses significantly improved water-holding capacity, dripping loss, MFI, SF, and springiness compared to higher doses of RNL. This result is consistent with the meat quality effects of natural herbs reported by others [6,56,57].

Notable differences in the initial carcass temperature between the experimental treatments may be attributed to the carcass preparation manner rather than to the treatments themselves. The defeathering process and traditional scalding used in this research comprised dipping the birds into a boiling tank, with potential temperature variation between different batches. The gap between batches was primarily due to the speed of the processing steps [56].

Meat quality has a number of characteristics, of which buyers are most concerned with color, tenderness, juiciness, taste intensity, and aroma [1]. The pH of the broiler meat is dependent on the quantity of glycogen in the muscle prior to slaughter and the rate of glycogen conversion to lactic acid after slaughter [9]. Our results were partially consistent with those of [58], who reported that the differences in pH in poultry meat could be ascribed to bird genotype, time, pre-slaughter stress, transfer, slaughtering manner, postmortem processing, storage duration, and the bird’s muscle type. This may reflect different pre-slaughter strategies for birds coping to stress, slaughter weights, and glycogen deposits at slaughter [59].

Meat color is both important for consumers and a simple indicator to investigate. Color changes are often the first signs of deterioration in quality and declining nutritional value. Fresh poultry meat should be light red in color. The cause of differences in the color of meat is due to pre-slaughter factors, stunning techniques, and cooling patterns [60], in addition to factors such as heme pigments, moisture content, sex, stress, protein status, and strain [61]. Postmortem heat and pH have an impact on protein degradation, which are reflected in the physical appearance by the quantity of reflected light from the inside and outside of the meat surface, indicative of the magnitude of protein denaturation [62]. In addition, the light dispersion affected the brightness (L*) of the meat, resulting in the concentration of heme pigment while having a minimal effect on the yellowness (b*) of the meat. Carotenoids improved the redness (a*) and yellowness (b*) of fresh meat, while Eimeria reduced them [63]. Infected/medicated groups that received nano-curcumin at 300 and 400 mg/kg doses had significantly higher a* values than the IUT group [10]. WHC reflects the capacity of meat to hold its own water, either in whole or in part [64]. WHC is among the most important criteria of meat quality and directly affects the sensory criteria, product yield, and the appearance of the end product [64]. WHC of the breast muscle is influenced by pre-slaughter conditions, processing techniques, and post-chilling conditions [65]. Eimeria infection reduced the WHC of un-supplemented chicken meat, but natural carotenoid supplementation improved it [63]. Here, WHC percentage decreased with increasing RNL dose. Increased water retention capacity of muscles boosts tenderness, juiciness, firmness, and appearance, which subsequently improves the quality of meat and monetary profit [9].

The muscle fiber size and the overall body fat content could change meat tenderness [59]. Tenderness appears to be the most important parameter for the quality of meat. It is subjectively assessed as a sense of firmness, elasticity, or flexibility [66]. Collagen solubility in cooking and the role of connective tissue in the movement of juice from muscle cells to the extracellular space may affect tenderness [67]. Juiciness is perceived as moisture and is a quality-determining trait. Higher juiciness in meat is related to adequate moisture [66]. Although the results of the shear strength and texture profile analyses demonstrated inconsistent trends for the supplementation with RNL powder, the inclusion of 1 g RNL powder demonstrated the best value of shear strength and hardness, and this was also true of springiness and cohesiveness. The higher the springiness, the more connective tissue, such as elastin and collagen, which results in the higher elasticity. This benefit to meat quality may be ascribed to the antioxidant properties of *Rumex nervosus* leaves, mainly flavonoids and phenolic compounds such as gallic acid present in RNL [25]. These compounds can delay oxidative reactions and hinder oxidation of lipids in meat to prevent spoilage resulting in rancidity and in unacceptable flavors and odors [68]. Generally, the tenderness of meat in response to the inclusion of RNL was increased. The quality of meat could indeed be enhanced by adding natural antioxidants, and *Rumex nervosus* has a strong antioxidant capacity due to its high phenolic content [22,23]. The authors of [69] investigated the natural antioxidant capacity of 26 spice extracts from 12 botanical families and found that many spices had high levels of phenolic content and high antioxidant capacity. In addition to RNL, cinnamon from Lauraceae, cloves from Myrtaceae, and oregano from Labiatae are possible sources of strong natural antioxidants for commercial use. According to [63], Eimeria infection did not reduce meat eating quality, natural carotenoids are effective antioxidants, and curcumin (300 mg/kg) fed alone or in combination with lutein was the most effective supplemented antioxidant compound. As a result, natural antioxidant supplementation of chicken feed may have desirable impacts on meat, as long as the quality of broiler meat did not demonstrate any harmful impacts upon consumption. RN is a good source of vitamin A and a poor source of vitamin E [20]. Vitamin A is formed from pro-vitamin A carotenoids, which are beneficial antioxidants and can enhance the color characteristics of animal products considered desirable by most consumers [70].

Broilers fed a diet supplemented with curcumin or a combination of phytogenics such as thymol, carvacrol, curcumin, and cinnamaldehyde showed higher yellow intensity in the meat [11]. In the present trial, the yellow intensity of the broilers fed with RNL was lower than those of negative and positive controls. The meat of birds supplemented with RNL at higher levels (3 and 5 g) had lower water retention, leading to greater drip loss compared to the controls. This may have occurred because the oxidative defense system of the muscle had a direct effect on these variables [64]. Natural herbs such as curcumin may have inhibited the oxidation of membrane lipids, increasing muscle cell integrity and causing water to become trapped in the flesh [71]. Myofibril protein denaturation profiles revealed a reduction in myosin and actin peaks in the non-supplemented natural carotenoid group [63]. Here, the highest cooking loss was observed in the SAL group and may have been related to the low pH value.

Excessive production of inflammatory mediators may also lead to oxidative damage [72]. Oxidation is a common process that alters pigments, fats, and proteins and reduces the shelf life of meat [73]. Coccidiosis in chickens negatively affects the quality and stability of meat color, reduces protein levels, and increases the amount of water between and within myofibrils and the cell membrane. This suggests that lower water retention capacity may be the result of increased protein oxidation [63]. Water retention capacity also depends on differences in proteolysis, cell shrinkage, and water mobilization into the extracellular space [11]. In addition, curcumin additives and commercial microencapsulated phytogenic supplements can increase the quality of the meat by increasing healthy polyunsaturated fatty acids and decreasing lipid peroxidation, increasing the lifespan of the meat [11].

Chemical and bioactive components of RN leaves were demonstrated in our previous study through proximate analysis, high-performance liquid chromatography (HPLC), and gas chromatography–mass spectrometry (GC-MS) [26]. The crude protein, raw fiber, ash, neutral detergent fiber, acid detergent fiber, and stored energy were detected at 13.63%, 8.24%, 18.01%, 20.21%, 15.48%, and 3273.31 kcal/kg respectively, indicative of its capability as a nutraceutical. In addition, gallic acid was the most important component (700 μg/g) identified in RNL extracts based on HPLC analysis. A total of 13 compounds were identified when the RNL was analyzed by the GC-MS.

Taken together, these results indicate that RNL could be used as a natural additive in the poultry industry to enhance the quality of meat. This is the first documentation to show that there are substantial changes in the quality of meat when RNL is fed to *Eimeria tenella*-infected broiler chickens. The physiochemical implications of these changes remain unclear. In any case, future research may build on the information provided herein, to gain further insight into the physiochemical properties of meat after RNL feeding of *Eimeria tenella*-infected birds. The processing of broiler meat is closely linked to its final pH, which is primarily determined by the quantity of glycogen in the muscles at the time of death. In this research, RNL was introduced into the chicken diet to attempt to enhance the quality of broiler meat; however, further research is required to elucidate the molecular mechanisms involved in the variation of these characteristics, as well as genetic determinism of glycogen levels and related meat quality traits.

## 5. Conclusions

In conclusion, RNL can be used to treat coccidia without adverse effects on the carcass traits or meat quality when compared to the non-infected, non-treated group. Dietary RNL powder supplementation at 1 g/kg was determined to have the most beneficial effects among RNL treatment groups with respect to MFI, reduced shear vigor, maximum springiness, and reduced pH of breast meat, and cooking loss. Dietary RNL powder supplementation led to reduced WHC percentage and higher dripping loss percentage. SAL treatment led to the highest CL, the best MFI, the highest hardness, and the lowest final pH. *Eimeria tenella*-infected broiler chickens at 2 weeks after infection exhibited higher relative pancreatic weights, reduced final luminosity (L*), and lower final whiteness indices in the meat compared to uninfected broiler chickens. The higher proportion of WHC was correlated with the higher percentage of drip loss with an increased dose of RNL. We conclude that birds that received 1 g RNL had better quality meat than those supplemented with 3 g RNL or 5 g RNL.

Based on the results obtained, it was concluded that broiler diets fortified with a RNL powder (5 g/kg) could enhance growth performance indices of *Eimeria tenella*-infected broiler chickens. However, broiler diets fortified with a small amount of RNL powder (1 g/kg) could elevate the quality and physicochemical attributes of meat. The infected groups had lower final brightness and higher relative pancreatic weight than the uninfected group. Conversely, the increased DL and shear force were RNL-dose-dependent. As a result, poultry diets supplemented with RNL powder could be a safe alternative to antibiotics and have numerous beneficial effects on consumer health and harmless effects on meat.

## Figures and Tables

**Table 1 animals-11-01551-t001:** Ingredients and calculated nutrients of broiler starter and finisher diets.

Ingredient	Period	
	Starter	Finisher
Yellow corn	53.218	58.09
Soybean meal	37.85	32.15
Wheat bran	2.00	2.2
Corn gluten meal	1.4	0
Choline chloride CL 60	0.05	0.05
Corn oil	1.5	4.2
Dicalcuim phosphate DCP	1.98	1.615
Ground limestone	0.9	0.79
Salt	0.400	0.30
DL-methionine	0.292	0.25
Lysine-HCL	0.21	0.105
Vitamin–mineral premix ^1^	0.200	0.200
Total	100	100
Metabolic energy, kcal/kg	3000	3200
Crude protein, %	23.0	20.0
Non phytate P, %	0.48	0.405
Calcium, %	0.96	0.81
d-lysine, %	1.28	1.06
Sulfur amino acids, %	0.95	0.83
Threonine, %	0.86	0.71

^1^ Vitamin–mineral premix contains the following per kg: vitamin A, 12,000,000 IU; vitamin D3, 5,000,000 IU; vitamin E, 80,000 IU; vitamin K3, 3200 mg; vitamin B1, 3200 mg; vitamin B_2_, 8600 mg; vitamin B_3_, 65,000 mg; pantothenic acid, 20,000 mg; vitamin B_6_, 4300 mg; biotin 220 mg; antioxidant (BHA + BHT), 50,000 mg; B_9_, 2200 mg; B_12_, 17 mg; copper, 16,000 mg; iodine, 1250 mg; iron, 20,000 mg; manganese, 120,000 mg; selenium, 300 mg; zinc, 110,000 mg.

**Table 2 animals-11-01551-t002:** Live body weight (BW), live body gain (BWG), cumulative feed intake (FI), feed conversion ratio (FCR), performance index (PI), and production efficiency factor (PEF) of birds with experimental diets during 1–14 dpi.

Parameter/TRT	UUT	IUT	SAL	1 g RNL	3 g RNL	5 g RNL	±SEM	*p*-Value
ABW (g)								
21 days (1 dpi)	779 ^c^	781 ^c^	892 ^a^	778 ^c^	809 ^bc^	857 ^ab^	10.298	0.0002
28 days (7 dpi)	1285 ^a^	1120 ^d^	1238 ^ab^	1112 ^d^	1136 ^cd^	1200 ^bc^	13.716	<0.001
34 days (14 dpi)	1875 ^a^	1604 ^e^	1849 ^ab^	1690 ^d^	1745 ^cd^	1797 ^bc^	18.890	<0.001
Gain (g)								
1 to 7 dpi	84.300 ^a^	56.46 ^b^	57.74 ^b^	55.68 ^b^	54.38 ^b^	57.20 ^b^	1.982	<0.001
8 to 14 dpi	84.400	69.12	87.26	82.60	87.00	85.40	2.135	0.1141
1 to 14 dpi	84.328 ^a^	62.78 ^c^	72.50 ^b^	69.13 ^bc^	70.69 ^b^	71.28 ^b^	1.501	0.0001
FI (g)								
1 to 7 dpi	118.36 ^a^	99.36 ^c^	104.36 ^b^	96.88 ^c^	96.46 ^c^	98.80 ^c^	1.510	<0.001
8 to 14 dpi	135.02 ^a^	125.00 ^b^	130.70 ^ab^	126.62 ^b^	129.48 ^ab^	135.08 ^a^	1.174	0.0434
1 to 14 dpi	126.68 ^a^	112.19 ^b^	117.52 ^b^	111.75 ^b^	112.97 ^b^	116.93 ^b^	1.317	0.0020
FCR (g)								
1 to 7 dpi	1.41 ^b^	1.81 ^a^	1.81 ^a^	1.77 ^a^	1.77 ^a^	1.74 ^a^	0.030	<0.001
8 to 14 dpi	1.61 ^b^	1.87 ^a^	1.54 ^b^	1.54 ^b^	1.50 ^b^	1.61 ^b^	0.029	0.0002
1 to 14 dpi	1.50 ^c^	1.84 ^a^	1.63 ^b^	1.63 ^b^	1.60 ^b^	1.66 ^b^	0.020	<0.001
FE								
1 to 7 dpi	0.71 ^a^	0.57 ^b^	0.55 ^b^	0.57 ^b^	0.56 ^b^	0.58 ^b^	0.013	0.0013
8 to 14 dpi	0.62 ^a^	0.55 ^b^	0.66 ^a^	0.65 ^a^	0.67 ^a^	0.63 ^a^	0.011	0.0085
1 to 14 dpi	0.67 ^a^	0.56 ^c^	0.62 ^b^	0.62 ^b^	0.62 ^b^	0.61 ^b^	0.008	0.0013
PEF								
1 to 7 dpi	339.13 ^a^	236.97 ^b^	254.19 ^b^	237.88 ^b^	237.49 ^b^	257.61 ^b^	7.629	<0.001
8 to 14 dpi	344.82 ^a^	259.87 ^b^	362.71 ^a^	324.37 ^a^	344.95 ^a^	334.88 ^a^	7.714	0.0001
1 to 14 dpi	341.97 ^a^	248.42 ^c^	308.45 ^b^	281.13 ^b^	291.22 ^b^	296.24 ^b^	6.365	<0.001
PI								
1 to 7 dpi	60.35 ^a^	33.14 ^b^	32.03 ^b^	32.36 ^b^	30.71 ^b^	33.29 ^b^	2.071	<0.001
8 to 14 dpi	52.96 ^a^	38.86 ^b^	59.15 ^a^	53.99 ^a^	58.69 ^a^	54.52 ^a^	1.604	0.0002
1 to 14 dpi	56.65 ^a^	36.00 ^c^	45.59 ^b^	43.17 ^b^	44.70 ^b^	43.90 ^b^	1.306	<0.001

^a–d^ Means (*n* = 25 birds per treatment) in same rows followed by different superscripts are significantly different (*p* < 0.05). Abbreviations: UUT, Negative control, un-supplemented, unchallenged; IUT, Positive control, un-supplemented, challenged; SAL: basal diet supplemented with anticoccidial salinomycin, challenged; *Rumex nervosus* leaves groups (basal diet supplemented with 1, 3, and 5 g RNL powder/kg diet respectively, challenged). SEM = standard error of the mean.

**Table 3 animals-11-01551-t003:** Slaughter variables of broilers supplemented with *Rumex nervosus* at 14 dpi.

Treatment	UUT	IUT	SAL	1 g RNL	3 g RNL	5 g RNL	SEM	Probability
Live weight (kg)	1759	1614	1747	1670	1797	1700	20.00	0.09
Carcass or dressing weight (kg)	1171	1052	1179	1109	1167	1129	14.39	0.07
Dressing yield %	66.58 ^a^	65.15 ^b^	67.44 ^a^	66.39 ^a^	66.56 ^a^	66.42 ^a^	0.192	0.02
^1^ Body components								
Heart %	0.45	0.46	0.46	0.48	0.50	0.55	0.011	0.06
Liver %	1.72 ^c^	1.95 ^ab^	1.90 ^ab^	2.00 ^a^	1.87 ^b^	1.95 ^ab^	0.020	<0.001
Proventriculus %	0.38	0.38	0.34	0.37	0.41	0.43	0.010	0.06
Gizzard %	2.04 ^b^	2.31 ^ab^	2.00 ^b^	2.08 ^b^	2.34 ^ab^	2.63 ^a^	0.064	0.02
Bursa %	0.21	0.16	0.21	0.22	0.20	0.20	0.007	0.33
Spleen %	0.08 ^c^	0.10 ^bc^	0.12 ^a^	0.11 ^ab^	0.09^bc^	0.13 ^a^	0.004	0.001
Thymus %	0.31	0.35	0.34	0.33	0.38	0.33	0.013	0.82
Breast %	26.55 ^bcd^	25.64 ^d^	28.34 ^a^	27.13 ^abc^	26.06 ^cd^	27.57 ^ab^	0.225	0.001
Leg %	20.4	19.97	19.4	20.16	19.86	18.83	0.251	0.547
Fat %	0.87	0.67	0.85	0.81	0.64	0.61	0.046	0.43
Pancreas %	0.18 ^b^	0.24 ^a^	0.27 ^a^	0.27 ^a^	0.24 ^a^	0.25 ^a^	0.008	0.001

^a–d^ Means (*n* = 5 birds per treatment) in the same row with different superscripts differ significantly (*p* < 0.05, 0.01, or 0.001). Abbreviations: UUT, negative control, un-supplemented, unchallenged; IUT, positive control, un-supplemented, challenged; SAL: basal diet supplemented with coccidiostat SAL, challenged; *Rumex nervosus* leaves groups (basal diet supplemented with 1, 3, and 5 g RNL powder/kg diet respectively, challenged). SEM, standard error of the mean. ^1^ Body components were measured as the ratio to the live weight.

**Table 4 animals-11-01551-t004:** The pH and color of breast meat (pectoralis major) were measured at 34 days of age in broiler chickens fed diets containing different levels of *Rumex nervosus* leaves (RNL), 24 hours (h) postmortem.

Treatment	UUT	IUT	SAL	1 g RNL	3 g RNL	5 g RNL	SEM	Probabilities
Initial pH, 1 h	6.08	6.16	6.23	6.26	6.28	6.17	0.034	0.637
Ultimate pH, 24 h	5.78 ^bc^	5.79 ^bc^	5.76 ^c^	5.86 ^ab^	5.76 ^c^	5.88 ^a^	0.013	0.010
pH Decrease	0.31	0.37	0.46	0.40	0.52	0.29	0.037	0.467
L*	49.91 ^a^	44.29 ^b^	43.18 ^b^	44.31 ^b^	43.01 ^b^	46.69 ^ab^	0.666	0.006
a*	7.25	7.94	7.55	5.97	7.87	8.56	0.413	0.629
b*	10.48	10.79	9.95	9.77	12.32	13.39	0.569	0.403
∆E	45.48 ^b^	51.09 ^a^	52.01 ^a^	50.77 ^a^	52.55 ^a^	49.38 ^ab^	0.686	0.018
Chroma	12.81	13.4	12.54	11.49	14.68	15.98	0.648	0.421
Hue angle	55.68	53.49	53.44	57.96	57.9	56.6	1.14	0.794
BI	33.82	41.13	38.71	35.91	46.65	47.55	2.322	0.468
WI	48.25 ^a^	42.66 ^b^	41.79 ^b^	43.01 ^b^	41.12 ^b^	44.25 ^ab^	0.690	0.021

The pH of the breast muscle was measured at 1 and 24 h postmortem, and color values were measured at 24 h. Abbreviations: UUT, negative control, un-supplemented, unchallenged; IUT, positive control, un-supplemented, challenged; SAL: basal diet supplemented with coccidiostat SAL, challenged; *Rumex nervosus* leaves groups (basal diet supplemented with 1, 3, and 5 g RNL powder/kg diet respectively, challenged). L*, lightness; a*, redness; b*, yellowness; ∆E, total color change; WI, whiteness index; hue, hue angle; BI, browning index; Chroma, saturation index; ^a, b, c^ Means (*n* = 5 birds per treatment) in the same rows followed by different superscripts significantly differ (*p* < 0.05, 0.01, or 0.001). SEM, standard error of the mean.

**Table 5 animals-11-01551-t005:** Water-holding capacity (WHC), cooking loss (CL), dripping loss (DL), myofibril fragmentation index (MFI), and shearing force (SF) in the breast meat of broilers fed experimental diets supplemented with *Rumex nervosus* leaves.

Treatment	UUT	IUT	SAL	1 g RNL	3 g RNL	5 g RNL	SEM	*p*-Value
WHC%	31.85 ^ab^	32.35 ^a^	30.38 ^abc^	26.94 ^d^	27.97 ^cd^	29.22 ^bcd^	0.494	0.0015
CL%	23.36 ^d^	22.46 ^d^	35.75 ^a^	31.21 ^b^	20.83 ^e^	25.92 ^c^	0.998	<0.001
DL%	1.55 ^b^	0.99 ^c^	1.01 ^c^	0.91 ^c^	2.04 ^a^	2.14 ^a^	0.104	<0.001
MFI%	112.85 ^bc^	104.80 ^c^	123.33 ^a^	118.03 ^ab^	57.90^e^	83.73 ^d^	4.346	<0.001
SF (kgf)	1.328 ^b^	1.404 ^ab^	1.470 ^ab^	0.86^c^	1.688 ^ab^	1.763 ^a^	0.0556	<0.001

Each mean is based on measurements from five breasts per treatment. ^a–d^ Means in the same row with different superscripts significantly differ (*p* < 0.05, 0.01, or 0.001). Abbreviations: UUT, negative control, un-supplemented, unchallenged; IUT, positive control, un-supplemented, challenged; SAL: basal diet supplemented with coccidiostat SAL, challenged; *Rumex nervosus* leaves groups (basal diet supplemented with 1, 3, and 5 g RNL powder/kg diet respectively, challenged).

**Table 6 animals-11-01551-t006:** Texture profile analysis in breast meat of broilers fed a diet supplemented with *Rumex nervosus* leaves.

Treatment	UUT	IUT	SAL	1 g RNL	3 g RNL	5 g RNL	SEM	Probability
Hardness	5.02 ^b^	6.08 ^b^	9.20 ^a^	6.63 ^b^	7.08 ^b^	6.63 ^b^	0.308	0.004
Springiness	0.91 ^a^	0.88 ^ab^	0.90 ^a^	0.90 ^a^	0.79 ^c^	0.80 ^bc^	0.013	0.007
Cohesiveness	0.43	0.44	0.45	0.41	0.40	0.44	0.007	0.129
Chewiness	2.07 ^b^	2.46 ^b^	3.67 ^a^	2.40 ^b^	2.29 ^b^	2.30 ^b^	0.128	0.004

Each mean is based on measurements from five breasts per treatment. ^a, b, c^ Means in the same row with different superscripts significantly differ (*p* < 0.05, 0.01, or 0.001). Abbreviations: UUT, negative control, un-supplemented, unchallenged; IUT, positive control, un-supplemented, challenged; SAL: basal diet supplemented with coccidiostat SAL, challenged; *Rumex nervosus* leaves groups (basal diet supplemented with 1, 3, and 5 g RNL powder/kg diet respectively, challenged).

## Data Availability

All datasets collected and analyzed during the current study are available from the corresponding author upon reasonable request.

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
