# Peer review of "Effect of Rumex nervosus Leaf Powder on the Breast Meat Quality, Carcass Traits, and Performance Indices of Eimeria tenella Oocyst-Infected Broiler Chickens"

_animals, 2021, doi:10.3390/ani11061551_

Round 1
Reviewer 1 Report
The revision corrected the required parts making the publication more fluid.
The paper can be published.
Yours sincerely
Author Response
Comments:
The revision corrected the required parts making the publication more fluid.
The paper can be published.
Yours sincerely
Response
Thank you for your time, your effort in reviewing our manuscript, and your recommendation that it be published.
Reviewer 2 Report
Authors have revised their work adequately. The only point yet that can be improved are comparisons with data from papers describing anticoccidial effect either in vivo or in vitro such as oregano or garlic bidens pilosa, sophora and artemisia.
Author Response
Thank you so much for reviewing our manuscript and adding insightful visions that made it more fluid and reading it's amazing.
Comments:
The authors have revised their work adequately. The only point yet that can be improved are comparisons with data from papers describing anticoccidial effect either in vivo or in vitro such as oregano or garlic bidens pilosa, sophora, and artemisia.
Response
Thank you so much for your insightful comment. As requested, the authors included comparisons with data from papers describing anticoccidial effects in vivo or in vitro, and highlighted them in bright green color as follows:
In Lines 546-562:
“The performance indices did not differ significantly between synthetic and natural anticoccidial products, consistent with the comparative anticoccidial study [30]. In accordance with [43], the poor performance of infected birds, whether medicated or not, as a result of the coccidial challenge is associated with nutrient malabsorption, decreased absorptive surface area, and inflammation. Natural growth promoters such as Thymus vulgaris and Rosmarinus officinalis may play an important role in alleviating coccidiosis symptoms in poultry [44]. In broilers reared without anticoccidial drugs, the combination of garlic and oregano essential oil (OEO) had a potent anticoccidial effect in vitro as well as a growth-promoting effect [45]. In a dose-dependent manner, graded doses of crude methanol extract of Garcinia kola significantly improved BWG in Eimeria tenella infected broiler chicks as a result of decreased fecal oocysts per gram [46]. In broiler chickens, Artemisia annua improves performance but has little anticoccidial activity [47]. Shi Ying Zi herbal powder could protect infected chickens from Eimeria tenella infection via prophylactic or therapeutic administration while also improving relative growth rate [29].”
In Lines 570-579:
“This could be because anticoccidial supplements, whether natural or synthetic, increased digestive enzyme activity and intestinal absorptive surface area [43]. They found that dietary OEO supplementation was undeniably more effective in terms of FCR. However, as a phytochemical with potential anticoccidial activity, OEO was less effective than the conventional in-feed agent monensin in promoting chicken growth after Eimeria tenella infection. Two weeks after Eimeria tenella infection, dietary OEO resulted in BWG and FCR that were comparable to the non-infected group [48]. OEO contains important bioactive components such as carvacrol and thymol [49].”
In Lines 605-610:
“Carcass characteristics like dressed yield percent and relative organ weight were higher in both infected and uninfected Madar leaf powder supplemented treatments, followed by amprolium and unsupplemented treatments [51]. Carcass characteristics of the infected/medicated nano-curcumin 300 mg/kg group improved when compared to the IUT and infected/medicated nano-curcumin 400 mg/kg groups [10]. “
In Lines 665-668:
“Carotenoids improved the redness (a*) and yellowness (b*) of fresh meat while Eimeria reduced it [65]. Infected/medicated groups that received nano curcumin at 300 and 400 mg/kg doses had significantly higher a* values than the IUT group [10].”
In Lines 674-675:
“Eimeria infection reduced the WHC of unsupplemented chicken meat, but natural carotenoid supplementation improved it [65]”.
In Lines 705-709:
“According to [65], Eimeria infection did not reduce meat eating quality, natural carotenoids are effective antioxidants, and curcumin (300 mg/kg) fed alone or in combination with lutein was the most effective supplemented antioxidant compound. ”
In Lines 726-728:
“Myofibril protein denaturation profiles revealed a reduction in myosin and actin peaks in the non-supplemented natural carotenoid group [65].”
Thank you so much for reviewing our manuscript and adding insightful visions that made it more fluid and reading its amazing.
Reviewer 3 Report
I understand that the 'highlighted' sections in the text of the manuscript provided by the authors correspond to modifications (inclusion/changes/corrections) made on the the original document. Also,
I noticed that part of this very same study has been published in "Animals". Taking this fact into consideration, I am inclined to accept the the justifications presented by the authors and consider that the study on Rumex nervosus should be published in order to complete the piece of information.
As mentioned above, I suppose the 'highlighted' text represents adjustments suggested by other reviewers. However, I feel obliged to declare that the 'highlighted' text needs extensive improvement in terms of sentences of difficult understanding, ideas not so clear and it seems that the language has not been properly revised (as in the first version - probably the rewritten part was made with some hurry)I understand that the 'highlighted' sections in the text of the manuscript provided by the authors correspond to modifications (inclusion/changes/corrections) made on the the original document. Also,
I noticed that part of this very same study has been published in "Animals". Taking this fact into consideration, I am inclined to accept the the justifications presented by the authors and consider that the study on Rumex nervosus should be published in order to complete the piece of information.
As mentioned above, I suppose the 'highlighted' text represents adjustments suggested by other reviewers. However, I feel obliged to declare that the 'highlighted' text needs extensive improvement in terms of sentences of difficult understanding, ideas not so clear and it seems that the language has not been properly revised (as in the first version - probably the rewritten part was made with some hurry)
Author Response
Thank you for your time and effort in reviewing our manuscript and providing us with all of your comments and suggestions.
Comments:
I understand that the 'highlighted' sections in the text of the manuscript provided by the authors correspond to modifications (inclusion/changes/corrections) made on the the original document. Also,
I noticed that part of this very same study has been published in "Animals". Taking this fact into consideration, I am inclined to accept the justifications presented by the authors and consider that the study on Rumex nervosus should be published in order to complete the piece of information.
As mentioned above, I suppose the 'highlighted' text represents adjustments suggested by other reviewers. However, I feel obliged to declare that the 'highlighted' text needs extensive improvement in terms of sentences of difficult understanding, ideas not so clear and it seems that the language has not been properly revised (as in the first version - probably the rewritten part was made with some hurry).
Response
Of course, this work is the continuation of a research already published in "Animals". Thank you so much for reviewing our manuscript and considering this fact. Thank you for accepting the authors' justifications and understanding that the Rumex nervosus study should be published in order to complete the information.
The authors revised the 'highlighted' text by simplifying difficult sentences and clarifying ambiguous concepts. In fact, because the rewritten section highlighted in yellow was completed in a hurry, the authors properly re-revised the language to match the first version. Moreover, highlight re-revised words or sentences within the text in bright green.
Thank you so much.
This manuscript is a resubmission of an earlier submission. The following is a list of the peer review reports and author responses from that submission.
Round 1
Reviewer 1 Report
Comments:
The work is well conducted and is the continuation of a research already published.
The purpose is clear, the results are probably affected by the low number of samples (only 5 per group)
In the Materials and methods, to be added:
composition Rumex nervosus Leaf Powder (also citing only publication 18)
Results, to be added:
after infection with Eimeria tenella, coccidiosis was present in broilers (also citing only publication 18)
Line 279 - remove the double parenthesis - infection (dpi)) -
Reviewer 2 Report
The authors have conducted an interesting and orginal data. However, they decide to provide data on meat quality and not on performance. However, this action makes manuscript less important, because its very important to know if the dosage that supports health, reduce intestinal lesions and can improve the meat quality. Now, all the introduction relates to health and anticoccidial effects, however, methodology, results and discussion are working with meat quality. Authors must change introdusction part.
Reviewer 3 Report
This study has a confusion of objectives and a faulty experimental design. Is the study intended to evaluate the effect of RNL on carcass quality, the effect of RNL upon infection with an Eimeria species, or the effects of salinomycin or coccidiosis upon carcass parameters?
Methods and Results: None of the Methods (or Results) are concerned with coccidiosis so why include Eimeria infection in this study? Almost all the Methods and the Results are concerned with meat and carcass quality etc. with a profusion of measurement details that are described in great and incomprehensible detail ( and also mostly statistically insignificant). This is unlikely to be of any interest to the general reader. It is suggested that a more appropriate Journal for this study would be one concerned directly with meat and carcass quality of livestock.
There is confusion regarding the diets underlying the six treatments. Thus, treatment groups 1,2, 3 are given a “basic” diet, whereas treatment 4 is given a “commercial” diet. We are not informed of the diet given to groups 5 and 6. Thus diets seem to differ regardless of whether birds are given RNL, salinomycin, or no supplement. This alone invalidates any comparisons because it is impossible to say whether the few differences found are due to treatments or these varied diets.
All groups except the negative control are infected with Eimeria. Appropriate controls would include uninfected birds given these treatments. It is not possible from this design to determine whether RNL at the various concentrations have any effect upon carcass parameters.
What is the point of infecting birds? Is this a coccidiosis study? If so then infecting birds at 21 days would be too late as experimental coccidiosis infection is usually carried out in much younger birds (what precautions were taken to prevent adventitious infection prior to 21 days (a common problem in experimental studies?). Was fecal material examined prior to this age to detect the presence of oocysts and as a consequence accidental infection? Why were classical criteria for measuring Eimeria infection (such as lesion scores) not carried out?
It is unclear why a group given salinomycin was included. Is this a drug evaluation study? Is this intended to evaluate the effects of infected birds given SAL upon carcass quality? If so, an uninfected group given the drug would be required.
The term “cocci” is slang and is not appropriate. Coccidiosis is the disease not the parasite.
In any study evaluating the effects of a supplement at relatively high concentrations it is important to include a “filler” of similar nutritional value to replace by volume the inclusion.
Line 23: Should be: Coccidiosis a disease caused by Eimeria.
Line 39-40: No evidence is provided that RNL had any effect upon the Eimeria infection. The conclusion that RNL improved carcass traits and meat quality in infected birds requires a control to establish that these findings do not occur in uninfected birds.
Line 47: You cannot challenge with eimeriosis.
Line 48: What is meant by “strengthening birds with RNL?
Line 51-53: It would be necessary to include a group of uninfected birds given RNL to conclude that the supplement was effective.
Line 51-55: The benefits reported in the conclusions are attributed to RNL with no mention of Eimeria infection. As already stated why include infected birds in the design?
Line 72-73: There is no evidence that “coccidiostats” cause bacterial resistance.
Line 78: The statement that “many of these drugs have been prohibited” is not true. Many anticoccidial agents are approved for use in the EU and elsewhere.
Line 82-83: There is no evidence that “forest products, shrubs and tea leaves” etc. are alternatives to “ionophore coccidiostats”. This is wishful thinking. The former are not employed in the modern poultry industry whereas the latter are used very widely.
Line 82-86: These sentences are unproven statements in an attempt to justify the present study. What are the “significant and appealing characteristics” of natural products? There is no evidence that unspecified natural products provide “higher production maintenance” than chemical additives. If this were so why is it that none are approved for use in Western poultry Industries? In this study there was no significant difference in final live weight of birds given RNL, the IUT or those given SAL.
The selective references quoted do not provide evidence that, in direct comparison, natural products are superior to existing drugs. It should be noted that efficacy of most natural products likely depends upon unspecified chemical ingredients that do not address such issues as toxicity and drug resistance. Few if any have undergone the necessary testing required by regulatory agencies.
Line 123-124: “The dose given is 4 x 104 cells –“. What number is this?
Line 137: There is no such word as “anticoccidiosis”.
Line 139-140: The description of the method for incorporation of RNL in the feed is inadequate and could not be repeated. SAL and RNL inclusion are accomplished by “mixed on top of the broiler fed”. This is not an approved method for incorporating any anticoccidial drug in the diet. Spell feed correctly.
Table 2: Only one of 13 slaughter variables is significant. This entire Table could simply be replaced by a sentence stating this fact. Why complicate the Table by providing three decimal places for non-significant data?
Line 392-402: This entire paragraph once again attempts to criticize antibiotic use in poultry. One such involves the statement that they pose a serious risk of residues. It should be pointed out that those antibiotics approved for use have been rigorously tested and are removed from the feed of broilers as long as two weeks prior to slaughter. There is no convincing published evidence to support the claim that residues are present or have caused a problem for consumers. If this were so, they would not be approved for use. It is notable that none of the so-called natural products have been tested to establish whether they also cause residues. They presumably contain pharmacologically active substances which in theory could also result in residues in poultry meat. It is not necessary for the authors to criticize antibiotics in order to justify this study.
Line 404-405: It is stated that “RNL reduces weight loss accompanying coccidiosis”. However, no evidence to support this is provided because the final live weights of the treatments were not significantly different (Table 2).
Reviewer 4 Report
It is my understanding that, in order to be published, a paper must bring an expansion to the knowledge on the topic, increase the robustness of previous findings, bring new insights to a research subject, validate a technology, among other requirements. I recognize the potential of Rumex nervosus leaf powder as a possible feed additive for chickens. However, I see limitations in the present manuscript that make me not recommend it for publication. I believe that the biggest limitation refers to sample size used for carcass and meat quality evaluations. Sample size for treatment means was 5 (1 chicken from each experimental unit) - considering the variations which are normally encontered among individuals, this sample size is far too small for allowing reliable results. To justify the statements above, I provide a few examples from the manuscript. First (relative to growth of the the chickens and not to the main focus of the report), the final average weight of the chickens in each treatment varied from 2816 grams to 1608 grams (Table 2). The p value of the analysis was 0.68, indicating non-significant differences. An experiment with enough power to detect important differences in broiler growth should be able to detect difference of 50 grams as significant. This means that the size of the experiment (5 cages of 5 chickens per treatment) did not provide reliability to the results. Also, the infected birds (positive control) performed better than the uninfected negative control, which is not in agreement with the hypothesis. Second, there is no reason to believe that the significant effect of treatments found for CL and DL (Table 4) are reliable, that is, changes from 20.83% to 35.75% in CL are to not likely to be due to treatments; likewise, changes from 0.99% to 2.14% in DL are also not likely and, in addition, the effects of treatments were not repeated for the two variables. These results are probably consequence of the elevated experimental error for the determination of these variables and the small sample size. Other examples could be pointed out to justify my considerations.